# Numerical Study on the Impact of Locked-In Stress on Rock Failure Processes and Energy Evolutions

**DOI:** 10.3390/ma16247519

**Published:** 2023-12-05

**Authors:** Tao Wang, Zishuo Liu, Liyuan Liu, Xianhui Feng

**Affiliations:** 1School of Civil and Resources Engineering, University of Science and Technology Beijing, Beijing 100083, China; tao.w@ustb.edu.cn (T.W.); m202120135@xs.ustb.edu.cn (Z.L.); fengxianhui@ustb.edu.cn (X.F.); 2Beijing Key Laboratory of Urban Underground Space Engineering, University of Science and Technology Beijing, Beijing 100083, China

**Keywords:** rock mechanics, locked-in stress, stress distribution, rock failure pattern, numerical simulation

## Abstract

Locked-in stress refers to internal stress present within rock formations that can influence the failure process of rocks under specific conditions. A simplified mechanical model is applied, drawing on elasticity and the hypothesis of locked-in stress, to explore the influence of locked-in stress on the mechanical properties of loaded rocks. An analytical solution is obtained for the stress distribution in a failure model of rocks that include locked-in stress. The findings demonstrate that the geometry and orientation of stress inclusions within the rock influence the initiation and propagation of cracks under the combined influence of locked-in stress and high-stress conditions. Moreover, the presence of locked-in stress substantially reduces the rock’s capacity to withstand maximum stress, thereby increasing its susceptibility to reaching a state of failure. The increase in closure stress leads to a significant increase in the magnitude of the maximum stress drop and radial strain variation within the rock, resulting in reduced strength and a shortened life of the ageing failure of the rock. In addition, the influence of stress inclusions on energy dissipation is investigated, and a novel relationship is established between the roughness coefficient of the rock structure surface and the angle of the rock failure surface. This relationship serves to characterize the linear dynamic strength properties of rock materials containing locked-in stress. This investigation not only advances the comprehension of stress distribution patterns and the effects of locked-in stress on rock failure patterns but also facilitates a more precise portrayal of the nonlinear features of alterations in the rock stress-strain curve under the influence of confined stress. These findings provide a solid theoretical foundation for ensuring the safety of excavations in various deep engineering projects.

## 1. Introduction

Coal rocks are intricate heterogeneous media composed of solid minerals, mineral grains, and micro-pores that are typically saturated with fluids or other non-homogeneous substances [1]. The concept of “locked-in stress” was introduced by Müller [2]. It pertains to the stress state inside a rock following the removal of external forces or the elimination of boundary constraints subsequent to disturbances, tectonic deformations, or other stress-inducing factors. Imbalance in internal stresses caused by external load changes during deep rock excavation can result in the release of confining stresses, which can lead to serious geological hazards [3,4], including gas protrusion [5], water burst [6], rock explosion [7,8], and elevation of the tunnel floor [9]. “Locked-in stress” and “creep” were identified as key rock traits in subsequent studies by Chen et al. [10] and Kie [11], which provided guidance for further research. Stress inclusions in rocks also play a role in seismicity [12]. Various experimental methods and techniques have been used to conduct in-depth studies on confining stresses. For instance, An et al. [13] calculated two residual stress elastic strain energy density fields within rocks using X-ray diffraction, while Sijing and Zhongjian [14] studied the effects of microcrack density and length on confining stresses and rock failure. Wang et al. [15] specifically analyzed the study of uncoordinated deformation and determined the self-balancing confining stresses generated by uncoordinated deformation. These studies not only revealed the mechanism and characteristics of confining stresses but also provided a theoretical basis for the study of rock mechanical properties.

Scholars have proposed specific calculation methods for confining stresses based on their understanding of their existence. Yue [16,17] gave a formula for the effect of closed stress inclusions on the surrounding rock and found that they can cause rock bursts in locally intact, brittle surrounding rocks when they are destroyed. He et al. [18] made progress in the study of the mechanism of closed stress and proposed some empirical formulas. Geng et al. [19] classified the stress forms of confining stresses in a generalized manner and summarized various types of confining stress calculation formulas. Xu et al. [20] simulated confining stresses in confining stress inclusions by similar simulation experiments and obtained theoretical equations for confining stresses. Liu et al. [21] evaluated the importance of confining stresses on the mechanical properties of rock materials with the help of fitting analysis of quantitative equations in similar simulation experiments. Wang et al. [22] measured and calculated the residual stresses in quartz veins of silica-bearing slate by the X-ray diffractometer method.

Locked-in stress within rocks has been extensively studied, particularly their implications in deep mining operations and their impact on geological hazards [23,24]. However, this study aims to deepen our understanding by specifically focusing on the influence of locked-in stress on the mechanical properties of rocks through computational modeling [25]. In light of the complexities associated with locked-in stress and its potential ramifications, this research endeavors to establish a comprehensive theoretical framework [26,27].

Our hypothesis centers around the premise that variations in rock parameters correspond intricately with alterations in locked-in stress, thereby directly affecting the mechanical behavior of rocks [28,29]. Through advanced computational modeling techniques, this study seeks to elucidate the interdependencies between rock parameters and locked-in stress, aiming to provide a predictive model for the mechanical response of rocks under varying stress conditions. This investigation primarily adopts a modeling approach, leveraging numerical simulations to explore the nuanced relationships between locked-in stress and rock properties. By structuring the study in this manner, we aim to provide a theoretical foundation that not only comprehensively explains locked-in stress phenomena but also offers valuable insights for optimizing deep mining practices and enhancing safety protocols.

## 2. Stress Distribution of Rock Fracture

### 2.1. Rock Load Model without Locked-In Stress

#### 2.1.1. Mechanical Model

In this study, a planar loading model is developed to investigate the stress distribution inside a circular inclusion of radius a, located at the center of a circle with radius b on an infinitely large rock. A coordinate system is established with the inclusion’s center as the origin and the axes parallel to the boundary. A uniform pressure *q*_1_/2 and tension *q*_1_/2 are applied to the top and bottom and left and right boundaries, respectively, while the inclusion is not subjected to any confined stress. Based on the disk compression test scheme proposed by Hondros [30], a testing configuration similar to the circular disk inside the rock in the figure was used, where the compressive stress was uniformly applied across the entire diameter after simplification. A series of strain gauges were used along the horizontal and vertical diameters of the disk to measure local strains, and the analytical solution for the stress distribution at any point inside the disk under uniform loading of an arc-shaped load in a plane problem was derived. The stress distribution along the loading diameter was obtained using the elastic-based solution, where local stresses were considered positive for tensile stresses. The stress distribution law of rocks was derived based on superposition using this experimental basis. The calculation method transforms the force on a 3-D surface into a force on a 2-D surface, as shown in Figure 1. The superposition process and the stress variation around the rock inclusions are shown in Figure 2.

The model is divided into two parts: the rock matrix and the inclusion. Assumptions are made for each part, including the uniform distribution and isotropy of the rock matrix, as well as its perfect elasticity with small deformation until the stress limit is exceeded. Additionally, there is no confined stress acting on the inclusion [31], and the deformation and failure of the inclusion itself are not considered. The superposition method is used to facilitate the verification of the calculation [32].

In order to convert a force problem in Cartesian coordinates to solve it in polar coordinates, we can use the basic theory of elasticity. Consider that the stress components *σ_x_*, *σ_y_*, and *τ_xy_* are known in Cartesian coordinates. We can then find the stress components *σ_r_*, *σ_α_*, and *τ_rα_* in polar coordinates.

We can extract a small triangular plate a_1_ with a thickness of 1 that contains the x-plane, y-plane, and z-plane from the rock mass, as shown in Figure 3a. The ab edge corresponds to the x-plane, the ac edge represents the y-plane, and the bc edge corresponds to the z-plane. The stresses on each surface are depicted in Figure 3b.

Let the length of the bc edge be *ds*, and the lengths of the ab and ac edges be *ds cosα* and *ds sinα*, respectively. Similarly, we can extract a small triangular plate a_2_ that contains the x-plane, y-plane, and z-plane with a thickness of 1. Using the elastic mechanics equations, can the stress components in Cartesian coordinates be related to those in polar coordinates? The transformation equations are as follows (1):(1)$$\varepsilon_{r}=(\sigma_{x}\cos^{2}\alpha +\sigma_{y}\sin^{2}\alpha +2\tau_{xy}\sin\alpha \cos\alpha )/E\;\varepsilon_{\alpha }=(\sigma_{x}\sin^{2}\alpha +\sigma_{y}\cos^{2}\alpha -2\tau_{xy}\sin\alpha \cos\alpha )/E\;\gamma_{r\alpha }=\left\{\left(\sigma_{x}-\sigma_{y}\right)\sin\alpha \cos\alpha +\tau_{xy}\left(\cos^{2}\alpha -\sin^{2}\alpha \right)\right\}/G$$
where *σ_x_* is the x coordinate in the direction of the stress component, *σ_y_* is the y coordinate direction stress component, *τ_xy_*(*τ_yx_*) is the shear stress in the vertical direction, *α* is the polar angle, *ε_r_* is the axial strain, *ε_α_* is the cyclic strain, *γ_rα_* is the tangential strain, *E* is the elastic modulus, and *G* is the modulus of rigidity.

#### 2.1.2. Analytical Solution of Stress Distribution without Locked-In Stress

The inclusion stresses *σ_r_*, *σ_α_*, and *τ_rα_* are correlated with the Cartesian components *σ_x_* and *σ_y_* by the known transformation relationships.
(2)$$\sigma_{r}=\frac{\sigma_{x}+\sigma_{y}}{2}+\frac{\sigma_{x}-\sigma_{y}}{2}\cos2\alpha \;\sigma_{\alpha }=\frac{\sigma_{x}+\sigma_{y}}{2}-\frac{\sigma_{x}-\sigma_{y}}{2}\cos2\alpha \;\tau_{r\alpha }=\frac{\sigma_{x}-\sigma_{y}}{2}\sin2\alpha$$

When the effect of closed stress is not considered, the boundary conditions for the force acting on point A can be expressed as *σ_x_* = *q*_1_, *σ_y_* = −*q*_1_, and *τ_xy_* = 0. By substituting these conditions into the formula, solving it, and applying the superposition method, we can derive the analytical solution for the strain value at any point within the rock sample without closed stress.
(3)$$\varepsilon_{r}=\{\frac{q_{1}}{2}\cos2\alpha \left(1-\frac{3a^{2}}{r^{2}}\right)\left(1-\frac{a^{2}}{r^{2}}\right)\}/E\;\varepsilon_{\alpha }=\{-\frac{q_{1}}{2}\cos2\alpha \left(1+\frac{3a^{4}}{r^{4}}\right)\}/E\;\gamma_{r\alpha }=\{-\frac{q_{1}}{2}\sin2\alpha \left(1+\frac{3a^{2}}{r^{2}}\right)\left(1-\frac{a^{2}}{r^{2}}\right)\}/G,$$
where *q*_1_ is the external forces, *a* is the inclusion radius, *r* is the distance from A point from the center of the circle, *α* is the polar angle, *E* is the elastic modulus, and *G* is the modulus of rigidity.

### 2.2. Rock Load Model with Locked-In Stress

#### 2.2.1. Mechanical Model

In this model, we consider the existence of a locked-in stress envelope at the center of a circle with radius *b*, under the same loading conditions as the previous model. The envelope has a radius of *a* and a uniform locked-in stress of *P*. The upper and lower boundaries of the rock are subjected to a uniform pressure *q*_2_/2, and the left and right boundaries are subjected to a uniform tension *q*_2_/2, as shown in Figure 4.

The model consists of two parts: the rock matrix and the locked-in stress envelope. The following assumptions are made for each part: (1) The rock matrix is continuous, with uniformly distributed constituent materials that are isotropic and perfectly elastic until the stress limit is exceeded. (2) The locked-in stress envelope is located far from the boundary of the rectangular rock mass and its deformation and failure are not considered. The locked-in stress is uniformly distributed within the envelope and does not affect the rock mass outside of it.

At this point, the force analysis at any point A on the disc is shown in Figure 5, where *α* represents the angle between point A and the *x*-axis. Due to the axisymmetric stress condition in the rock, the stress component maintains a consistent numerical value at every point along a specific circumferential line, assuming that there exists a *z*-axis, perpendicular to the oxy plane. Therefore, the stress component symmetric around the *z*-axis in the polar coordinate plane is only a function of r and does not vary with *α*. The shear stress *τ_rα_* is 0. That is, only the stress-strain situation in the oxy plane is analyzed.

#### 2.2.2. Analytical Solution of Stress Distribution with Locked-In Stress

The analytical solution of the stress value at any point A of the rock under the joint action of the enclosed stress and the external force is obtained after considering the displacement boundary conditions:(4)$$\sigma_{r}=-\frac{\frac{b^{2}}{r^{2}}-1}{\frac{b^{2}}{a^{2}}-1}P-\frac{1-\frac{a^{2}}{r^{2}}}{1-\frac{r^{2}}{b^{2}}}q_{2}\;\sigma_{\alpha }=\frac{\frac{b^{2}}{r^{2}}+1}{\frac{b^{2}}{a^{2}}-1}P-\frac{1+\frac{a^{2}}{r^{2}}}{1-\frac{r^{2}}{b^{2}}}q_{2},$$
where *P* is locked-in stress, *q*_2_ is the external forces, *b* is the radius of the disk, *r* is the distance from A point from the center of the circle, and *α* is the polar angle.

According to the aforementioned calculations and in conjunction with the superposition method, the analytical solution for the closed stress in the rock subjected to external forces can be obtained, as shown in Equation (5).
(5)$$\varepsilon_{r}=\{\frac{q_{2}}{2}\cos2\alpha \left(1-\frac{3a^{2}}{r^{2}}\right)\left(1-\frac{a^{2}}{r^{2}}\right)-\frac{\frac{b^{2}}{r^{2}}-1}{\frac{b^{2}}{a^{2}}-1}P-\frac{1-\frac{a^{2}}{r^{2}}}{1-\frac{r^{2}}{b^{2}}}\frac{q_{2}}{2}\}/E\;\varepsilon_{\alpha }=\{-\frac{q_{2}}{2}\cos2\alpha \left(1+\frac{3a^{4}}{r^{4}}\right)+\frac{\frac{b^{2}}{r^{2}}+1}{\frac{b^{2}}{a^{2}}-1}P-\frac{1+\frac{a^{2}}{r^{2}}}{1-\frac{r^{2}}{b^{2}}}\frac{q_{2}}{2}\}/E\;\gamma_{r\alpha }=\{-\frac{q_{2}}{2}\sin2\alpha \left(1+\frac{3a^{2}}{r^{2}}\right)\left(1-\frac{a^{2}}{r^{2}}\right)\}/G,$$
where *P* is locked-in stress, *q*_2_ is the external forces, *b* is the radius of the disk, *a* is inclusion radius, *r* is the distance from A point from the center of the circle, *σ_r_* is radial stress, *σ_α_* is circumferential stress, *τ_rα_* is tangential stress, *α* is the polar angle, *E* is the elastic modulus, and *G* is the modulus of rigidity.

The analysis results in relatively compact formulae for the stress components inside the inclusion and their transfer on the boundary line of inclusion and matrix. There is the expectation that the clearly structured derivation supports the application in practice.

## 3. The Impact of Stress Inclusions on Rock Deformation and Failure

### 3.1. Analytical Method and Finite Element Method for Stress Inclusions Exist

To investigate the influence of inclusion type and geometry on the stress field distribution in the rock matrix and eliminate the effect of inclusion size on rock properties, uniaxial compression numerical simulation experiments were conducted in RFPA on a 100 mm high and 50 mm diameter cylindrical sandstone specimen (sandstone with a density of 2800 kg/m^3^, Poisson’s ratio of 0.25) containing several types of stress inclusions, with a press speed of 0.2 mm/s. RFPA Basic V2.0 is a rock failure process analysis software that is based on finite element theory and statistical failure theory for numerical simulation of material failure. The models in this thesis are modeled using the finite element numerical method, the elastic intrinsic law is used, the failure criterion is the Mohr-Coulomb yield criterion, and the mesh size is 0.5 mm × 0.5 mm. Figure 6 depicts the elastic damage constitutive relationship of the software’s microscale unit under uniaxial tension and compression. Here, the damage threshold criterion is defined by Equation (6), recognized as the Mohr-Coulomb criterion and the maximum tensile strength principle, along with Equations (7)–(9), representing the tensile and compressive parameters. Unit failure occurs when the simulated unit satisfies either of these equations, in accordance with the established criteria. The formula involves the damage variable, where *D* = 0 represents an undamaged state, *D* = 1 corresponds to a failure state, and 0 < *D* < 1 denotes varying degrees of damage; *λ* represents the ratio of ultimate stress to residual stress in tension or compression, and 0 < *λ* < 1.
(6)$$F_{1}=\sigma_{1}-\frac{1+\sin\varphi }{1-\sin\varphi }\sigma_{3}+f_{c}=0\;or\;F_{2}={-\sigma }_{3}+f_{t}=0$$
(7)$$E=(1-D)E_{0}$$
(8)$$D=\left\{\begin{array}{c} \;\;\;\;\;\;\;\;\;0,\;\;0_{}<\varepsilon <\varepsilon_{t0}\\ 1-\frac{\lambda \varepsilon_{t0}}{\varepsilon },\;\;\varepsilon_{t0}<\varepsilon <\varepsilon_{tu}\\ \;\;1,\;\;\varepsilon_{}>\varepsilon_{tu} \end{array}\right.$$
(9)$$D=\left\{\begin{array}{c} \;\;\;\;\;\;\;\;\;\;\;\;0,\;\;\varepsilon_{}\geq \varepsilon_{c0}\\ 1-\frac{\lambda \varepsilon_{c0}}{\varepsilon },\;\;\varepsilon_{}<\varepsilon_{c0} \end{array}\right.$$
where φ is the element friction angle, $f_{c}$ is the element of compressive strength, $f_{t}$ is the element tensile strength, $\sigma_{1}$ and $\sigma_{3}$ represent the maximum and minimum principal stresses of the element, respectively, *λ* is the element residual strength coefficient, *E*_0_ and *E* are the elastic modulus both before and after the initiation of damage, $\varepsilon_{t0}$ is the maximum tensile strain, and $\varepsilon_{c0}$ is the maximum compressive strain.

Figure 7 presents a comparison between the analytical method (AM) and the finite element method (FEM) for the stresses in the rock surrounding a circular hole parallel to the *x*-axis during the failure with stress inclusions. The stress state of the rock at the same strain is selected for verification to eliminate the influence of external forces. In both methods, the stress at the edge of the circular hole gradually decreases as the stress within the inclusions increases. However, at an internal stress of 4 MPa, the stress at the edge of the hole experiences a sudden and significant increase beyond the analytical solution. This abrupt increase indicates that the rock is approaching its tensile strength, resulting in failure at the edge of the hole due to the effect of stress closure. Consequently, the rocks’ locked-in stress affects the stress distribution around the hole. Moreover, when the rock’s locked-in stress reaches 3 MPa, the rock experiences failure due to the locked-in stress. These findings confirm that the increasing locked-in stress does indeed influence the stress state of the rock. By comparing the stress distributions obtained from the two methods under different confining stress conditions, the validity of the rock failure model equation in the presence of confining stress is demonstrated.

### 3.2. Rock Failure Processes According to Different Stress Inclusions Geometry

#### 3.2.1. Influence of Geometric Shapes of Stress Inclusions

After confirming the influence of stress inclusions on stress distribution in rocks, we proceed to investigate the impact of various types of stress inclusions on the stress field and mechanical properties of the rock matrix. This analysis takes into account different morphological and physical properties. The RFPA-simulated failure cracks were observed and analyzed. In Figure 8, following the maximum principal stress criterion, we observe that in rocks without internal stresses but containing circular inclusions, the maximum principal stress during the failure process concentrates in the surrounding rock near the inclusion, with a failure angle of approximately 90 degrees. The failure occurred along the direction of the maximum principal stress. As the internal stress within the inclusions increased, a local equilibrium field gradually formed between the inclusions and the compressive stress, leading to the initiation of vertical failure cracks. The angle of the terminated crack varied with the increase of internal stress, while the angle of the maximum principal stress remained relatively constant, with negligible reduction beyond this angle. On the contrary, rocks containing elliptical inclusions exhibited a higher tendency for stress concentration due to structural and stress differences. As the locked-in stress increased from 0 MPa to 6 MPa, the crack extended in both directions from the center, and the termination crack continuously changed with the different locations of the locked-in stress. Once the locked-in stress within the stress inclusions reached the tensile strength, internal and external cracks appeared simultaneously, significantly reducing the load-carrying capacity of the rock.

The rock undergoes compressive deformation until it reaches its tensile strength, ultimately resulting in fracture along the initial failure surface. In the presence of stress inclusions, rocks generally fracture along the direction of the maximum principal stress when subjected to both overlying rock pressure and stress inclusions. This leads to the formation of a failure fracture and a V-shaped structure along the free side of the excavation. During rock tunnel excavation, when the excavation is near the stress inclusions, the fragmented rock between the upper and lower sections loses its connection with the surrounding rock near the excavation space. As a result, it is displaced into the excavation space due to the combined influence of the overlying rock and stress inclusions, leading to a dynamic catastrophe. The influence of confining stress during rock loading is further confirmed by comparing numerical simulations of V-shaped rocks and rocks with stress inclusions, which are frequently encountered in rock tunnel construction.

#### 3.2.2. Influence of Stress Confinement Angle

The morphology and physical attributes of stress inclusions within a rock mass wield a significant influence over the mechanical properties of the rock matrix, affecting alterations in fracture behavior and the surfaces of fracture failures. To explore this influence, we conducted experiments examining the failure characteristics and morphology of sandstones with varying lamination angles. These experiments aimed to compare the failure behavior of sandstones containing stress inclusions under loading. Our compression experiments uncovered a critical impact of the inclusion angle on the distribution of crack morphology, often resulting in initiation cracks forming at the center of the specimen.

The stress concentration patterns depicted in Figure 9, Figure 10, Figure 11 and Figure 12 reveal that stress concentrations occur on both sides of the inclusions when the angle is 0°. Cracks that deviate from the vertical direction are observed primarily near or above the tensile strength threshold. As the angle increases to 30°, 60°, and 90°, stress concentrations predominantly manifest along the inclined angle of the stress inclusions, ultimately leading to the formation of vertical cracks and consequent failure. Moreover, the cracks exhibit diverse shapes in different types of inclusions, further highlighting the stochastic nature of rock failure cracks. Additionally, it is worth noting that as the stress within the inclusions intensifies, the residual stress upon crack formation increases, consequently increasing the likelihood of rock burst accidents.

Furthermore, microcracks may sprout and develop randomly in rock bodies with stress inclusions under the conditions of uniform stress distribution or no significant stress concentration. The sprouting mode and location are contingent on the microstructural variations within distinct tissues, resulting in diverse sprouting patterns. Stress concentrations strongly influence the location of fatigue crack sprouting, with cracks likely to occur in stress inclusions with large stress concentrations or at the edge of the matrix. In summary, the shape and angle of stress inclusions and material properties strongly influence the rock failure pattern.

### 3.3. The Impact of Stress Inclusions on Rock Peak Stress and Peak Strain

By comparing the stress-strain curves of standard cylindrical specimens with stress inclusions at different angles in Figure 13 and Figure 14, it is evident that the presence of stress inclusions causes a significant decrease in the maximum stress that the rock can withstand. Specifically, the maximum stress of the rock without inclusions is 77.3 MPa, while the maximum stress for elliptical inclusions without locked-in stress in the model ranges from 41.66 MPa for inclusions at 90° to 52.72 MPa for inclusions at 0°. The angular dimensions of the stress inclusions exhibit a correlation with the magnitude of stress and strain during uniaxial compression failure of rocks in the presence of such inclusions. Moreover, the shape of the stress inclusions influences the fracture behavior of the rock. Specifically, elliptical inclusions are prone to fracture along the long axis due to stress concentration, whereas 90° inclusions align more closely with the uniaxial compression failure pattern of the rock. After examining the stress distribution pattern along the *x*-axis, it was found that the stress distribution trend and stress concentration points are similar to those found in elastic materials. Stress concentrations have been found to increase for elliptical cavities and inclusions with increasing aspect ratios. The stress concentration at the edges of circular cavities has also been verified [34]. It is further argued that elliptical stress inclusions at 90° are more susceptible to cracking under a combination of external and internal pressures, which in turn leads to a decrease in the maximum stress that the rock can withstand and more rapid failure [35,36].

Furthermore, Figure 15 reveals that the presence of relatively small confining stresses within the rock pores leads to a slight increase in its stress-bearing capacity. Specifically, when confining stresses significantly below the tensile strength are present, they enhance the rigidity of rock crevices. However, as the confining stress reaches 6 MPa after surpassing the rock’s tensile strength, the maximum stress capacity of the rock undergoes a substantial decrease. Therefore, rock fracturing in the presence of stress inclusions is considered to be similar to hydraulic fracturing of prefabricated defective rocks, corroborating with experimental conclusions of hydraulic fracturing of granite to better understand the mechanisms of fracture initiation, extension, and coalescence in the presence of stress inclusions in rocks [37].

The study of stress nephogram changes within the rock and the analysis of microscopic fracture surface morphology revealed that the progressive deterioration of brittle rocks primarily arises from the propagation and enlargement of cracks, resulting in an ongoing decline in the mechanical characteristics of the rock. The Joint Roughness Coefficient (JRC), which represents the roughness of the rock’s structural surface, assumes a pivotal role in the investigation of rock compressive strength [38,39]. By employing the standardized contour curve method proposed by Barton and Nicholas, the JRC coefficients for various stress inclusion scenarios were computed to characterize the compressive strength of rocks under different conditions. As illustrated in Figure 16, for the failure of a rock containing 30° stress inclusions, it can be observed that the JRC coefficient decreases with increasing stress within the inclusions. This signifies a reduction in the roughness of the fracture surface following rock failure, indicating a gradual decline in the rock’s fracture strength.

It is essential to emphasize that rock failure can manifest in various forms, including shear or tensile-shear fractures, depending on the nature of the rock and whether it features homogeneous, compact, and robust structures or exhibits weaker rock failure mechanisms. The deformation patterns and fracture mechanisms in rock failure are influenced not only by lithology and stress levels but also primarily by the rock’s inherent structure and the presence of stress inclusions within it.

## 4. Discussion

Through a comparative analysis of theoretical models, it has been demonstrated that the presence of stress inclusions has a profound influence on stress distribution within rocks. This influence extends beyond external forces and encompasses variations in pole diameter, stress inclusion radius, disc radius, and pole angle. The derived stress equations reveal periodic relationships among different parameters. These correlations provide further validation for the mechanical model of rock splitting incorporating stress inclusions.

From an energy perspective, Qian [40] proposed a quantitative prediction principle for rock burst and impact pressure offers valuable insights into establishing strain-based criteria for rock burst classification based on energy principles. In dynamic failure scenarios, the remaining energy is transformed into kinetic energy. Therefore, the kinetic energy of the fragmented rock mass is approximately equal to the difference between the strain energy stored in the highly stressed surrounding rock and the dissipated energy during rock rupture. This principle facilitates the classification of rock bursts based on the amount of generated kinetic energy: no rock burst, slight rock burst, medium rock burst, and intense rock burst [41]. The derived quantitative prediction formula for rock burst energy, considering energy release and the conversion ratio of remaining energy into kinetic energy, provides a more intuitive measure of the kinetic energy associated with rock failure caused by the presence of rock inclusions. As depicted in Figure 17, with increasing stress within the inclusions, the magnitude of the rock rupture moment generated by the remaining kinetic energy becomes greater, indicating a heightened severity of rock explosion accidents. The presence of stress inclusions in rock failure increases the likelihood of occurrence, and as stress levels rise, the failure process becomes accompanied by more severe rock bursts and other related accidents.

Through the analysis of diverse rock failure scenarios, it has been observed that the presence of stress inclusions and stress levels significantly influence rock behavior. During the pre-compression stage, where stress remains below the threshold for crack initiation, the effect of heightened locked-in stress on rock failure characteristics is minimal, resulting in no significant failure (see Figure 18a,b). In the stable crack development stage, microcracks gradually form and expand under loading, concurrent with a continuous rise in internal stress within the rock (depicted in Figure 18c,d). Stress-induced microcracks create additional tension under constrained stress, leading to crack propagation and the potential sliding or frictional movement of extending or closing tensile cracks, ultimately culminating in rock failure (see Figure 10c and Figure 18d). Consequently, rocks experience a higher rate of stress reduction under locked-in stress conditions, with a more noticeable decrease in maximum stress. As stress levels exceed the compressive stress of the rock, microcrack development enters an unstable stage, and under high-stress conditions, locked-in stress enhances the formation of microcracks, resulting in an increased number of microcracks that gradually expand and connect to form a main crack. This process accelerates rock deformation and signifies impending rock destabilization. In real-world scenarios, prolonged exposure to locked-in stress leads to increased internal failure development and the progressive expansion of the main crack, making rocks more susceptible to rock burst incidents and other problems [42,43]. From this analysis, it can be concluded that the stress evolution characteristics of brittle sandstone under the coupling of locked-in stress and external stress are primarily determined by the effective stress level and the development state of cracks influenced by stress inclusions. When the effective stress surpasses the threshold for crack initiation, locked-in stress promotes crack development, ultimately resulting in stress release and a loss of rock strength, leading to instantaneous failure.

## 5. Conclusions

(1) A simplified mechanical model is applied in this study to investigate the deformation and failure behavior of rocks containing stress inclusions when subjected to external loading. The model is based on elasticity and incorporates the hypothesis of stress inclusions. It considers the combined influence of stress inclusions and external forces on rocks, and an equation describing the stress distribution in rocks containing stress inclusions under loading is derived. By providing insights into the stress distribution characteristics and the numerical validation based on finite element analysis, this model contributes to the theoretical understanding of stress distribution in deep engineering projects and the analysis of excavation safety.

(2) The presence of stress inclusions, along with locked-in stress, has a profound impact on the formation and propagation of cracks in rocks significantly reducing the maximum stress capacity of the rock and increasing the risk of failure. The application of locked-in stress results in a notable increase in the maximum stress drop and radial strain alteration within the rock. The rock weakens, accelerating the aging failure process. The emergence of a discernible “step” changes in the rock’s maximum stress, attributed to locked-in stress variations, indicates the growth and interconnection of microcrack failure. This makes the rock more prone to failure and potential engineering incidents. Consequently, it is crucial to prioritize safety concerns associated with locked-in stress in practical projects.

(3) The stress-induced failure process in typical sandstone under the combined influence of confining stress and high stress entails the evolution of the radial strain field, starting from uniform deformation and progressing towards localized microcrack initiation, extension, and eventual destabilization, leading to the formation of cracks at specific angles. However, the presence of stress inclusions introduces a notable delay in the development of localized zones within the axial strain field compared to the radial strain field. Additionally, the development of cracks within the shear strain field is not prominently observed.

(4) The Joint Roughness Coefficient (JRC) serves as a significant indicator of rock compressive strength, particularly in cases involving stress inclusions. Investigations indicate a decreasing trend in the JRC coefficient during rock failure, corresponding to the increase in stress within 30° stress inclusions. This signifies that stress inclusions impact the rock’s failure strength, with the reduction in post-fracture surface roughness reflecting the gradual decline in the rock’s failure strength.

Crack initiation typically transpires at locations where stress concentration is most pronounced within the inclusions when locked-in stress is minimal. As stress levels increase, the rock first undergoes tensile failure, followed by the combined influence of locked-in stress and pressure, resulting in rapid crack formation. From an energetic standpoint, the presence of locked-in stress leads to reduced energy dissipation during crack propagation. Consequently, the elastic strain energy accumulated within the rupture zone of the rock surpasses the total energy consumed by primary fracture and secondary crack expansion, leading to the sudden release of the remaining elastic strain energy and triggering rock burst incidents. Hence, the presence of stress inclusions impacts microcrack evolution, subsequently influencing the stress relaxation process in rocks and expediting their failure when subjected to external forces.

## Figures and Tables

**Figure 1 materials-16-07519-f001:**
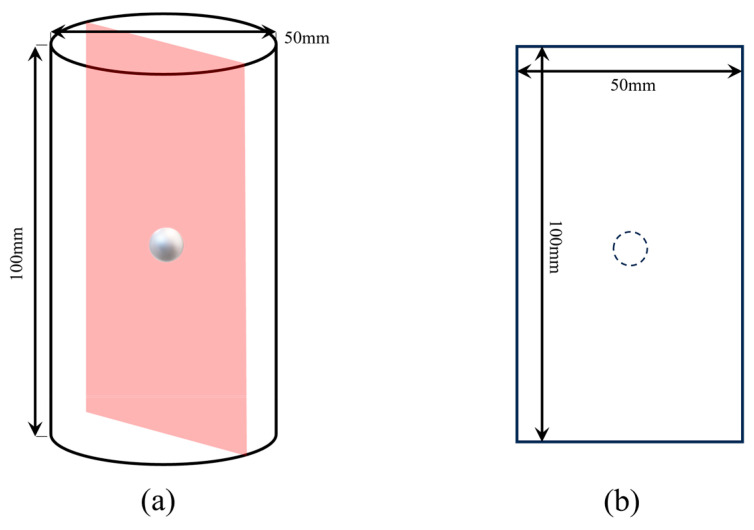
Simulation model. (**a**) Rock sample; (**b**) 2-D model.

**Figure 2 materials-16-07519-f002:**
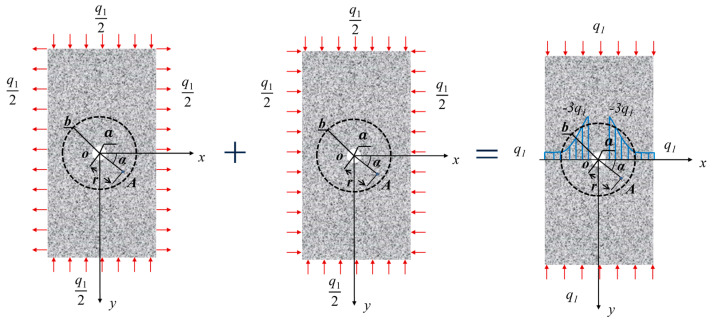
Force diagram of a rock without locked-in stress.

**Figure 3 materials-16-07519-f003:**
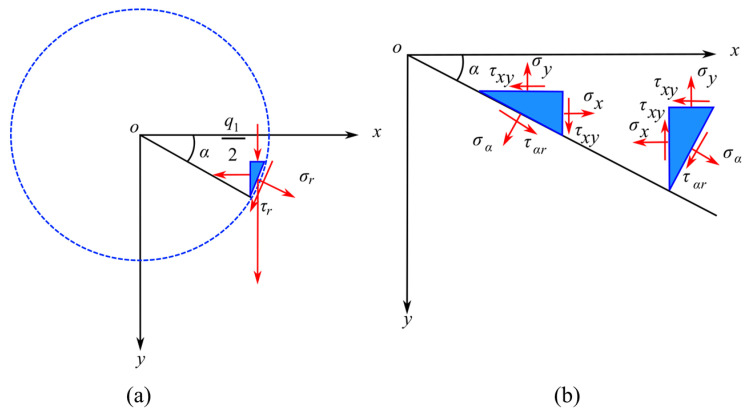
Stress analysis of point A on a disc without locked-in stress. (**a**) Macroscopic force acting on point A within the disc. (**b**) Force decomposition at point A within the disc.

**Figure 4 materials-16-07519-f004:**
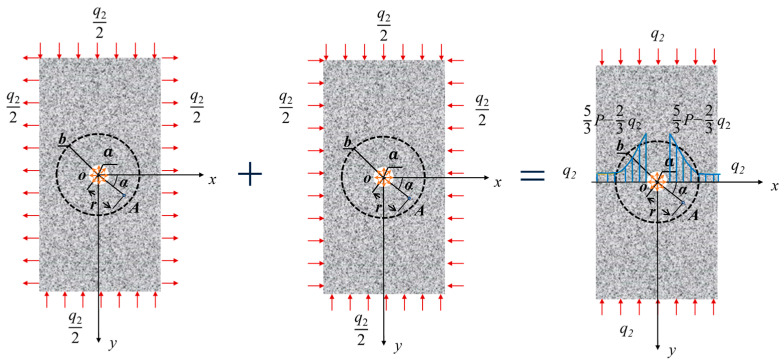
Force diagram of a rock with locked-in stress.

**Figure 5 materials-16-07519-f005:**
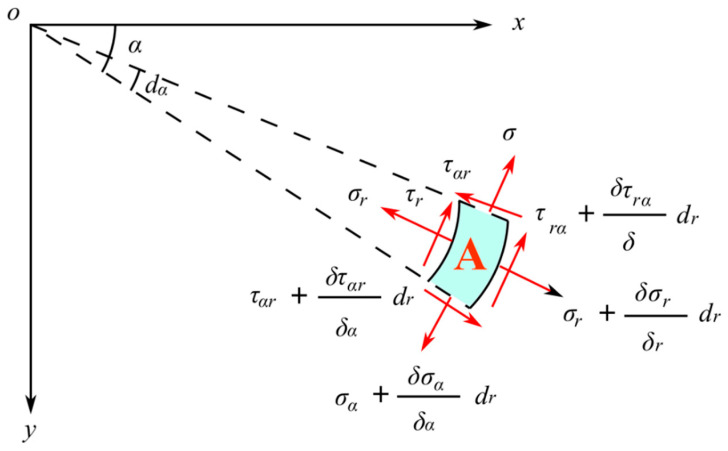
Stress analysis of a point A on a disc with locked-in stress.

**Figure 6 materials-16-07519-f006:**
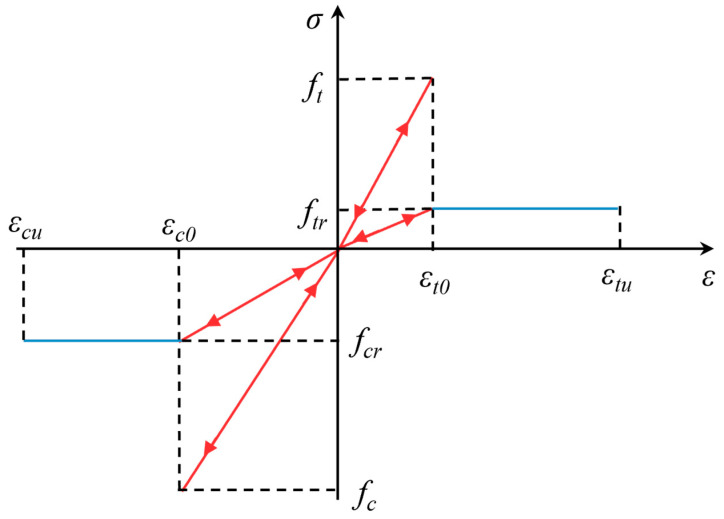
The elastic damage constitutive relationship of micro-scale units under uniaxial tension and compression [29,33].

**Figure 7 materials-16-07519-f007:**
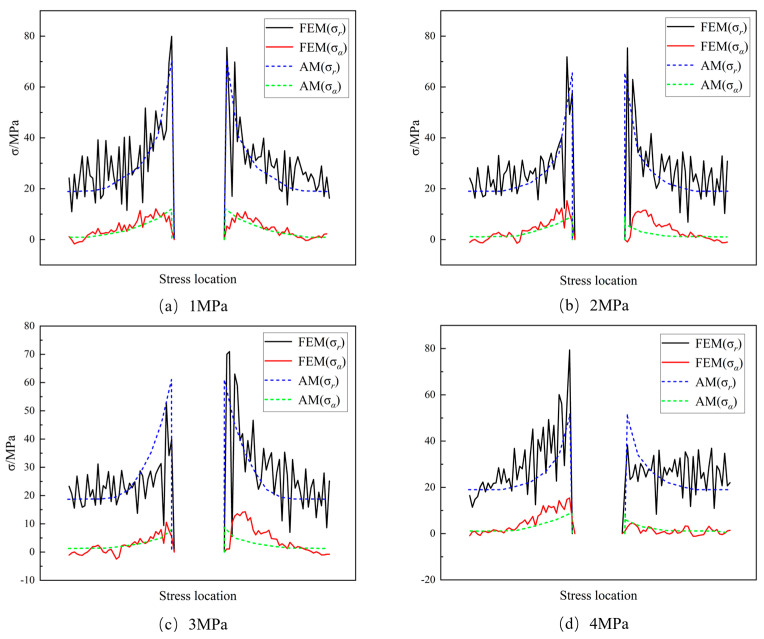
Stress state at the center of the rock circle for both methods.

**Figure 8 materials-16-07519-f008:**
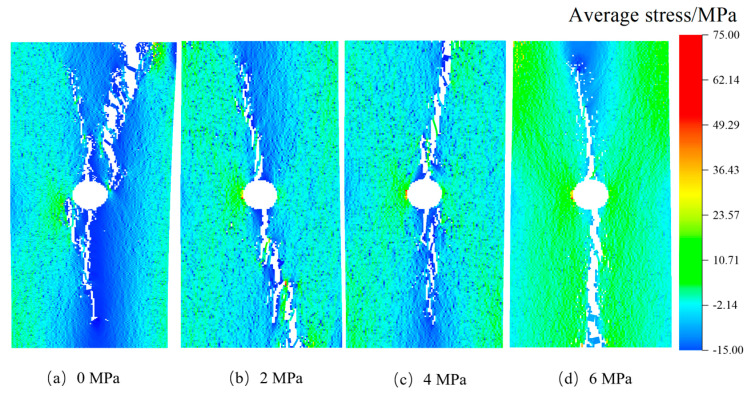
Cracking rock patterns with circle stress inclusion under different stresses.

**Figure 9 materials-16-07519-f009:**
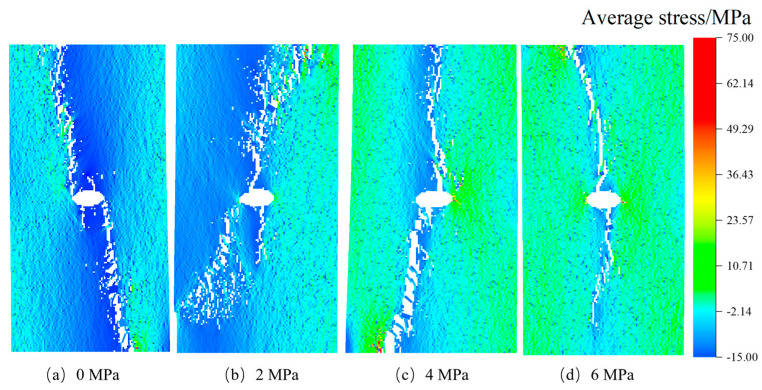
Cracking rock patterns with elliptical stress inclusion under different stresses.

**Figure 10 materials-16-07519-f010:**
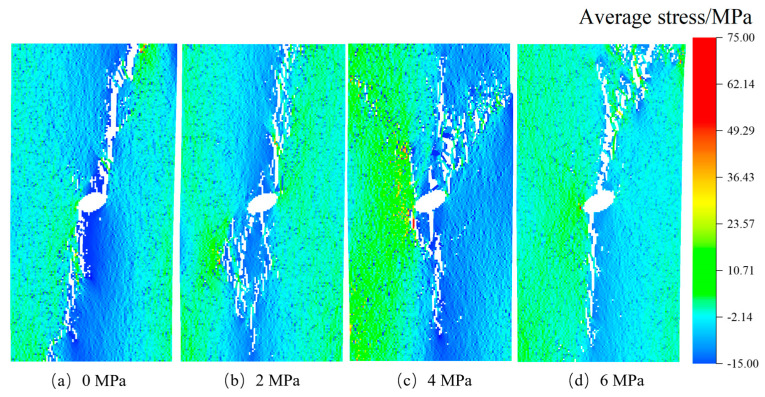
Cracking rock patterns with 30° elliptical stress inclusion under different stresses.

**Figure 11 materials-16-07519-f011:**
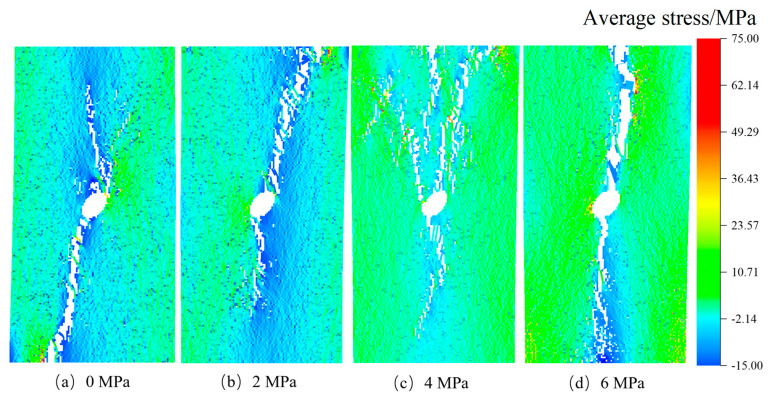
Cracking rock patterns with 60° elliptical stress inclusion under different stresses.

**Figure 12 materials-16-07519-f012:**
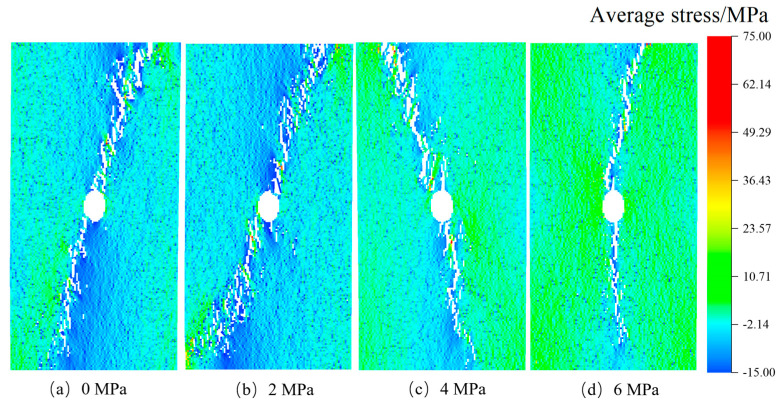
Cracking rock patterns with 90° elliptical stress inclusion under different stresses.

**Figure 13 materials-16-07519-f013:**
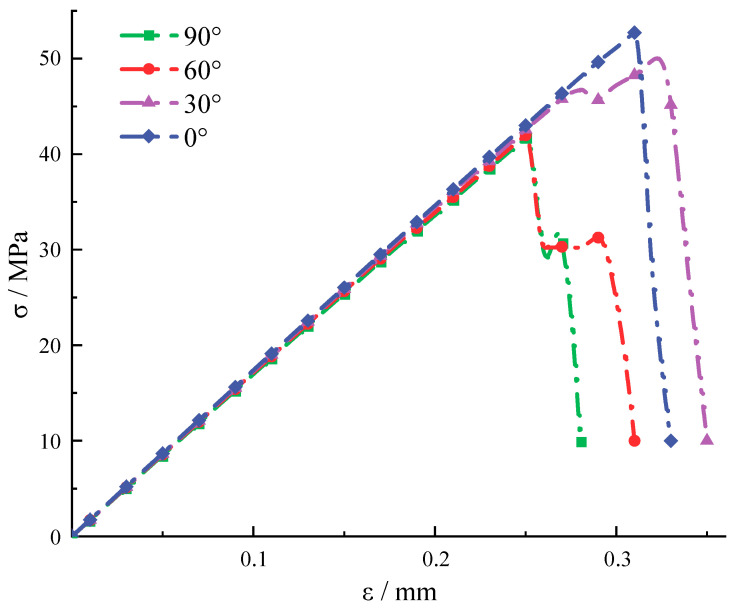
The whole process of stress-strain in rocks containing elliptical stress inclusions with different angles.

**Figure 14 materials-16-07519-f014:**
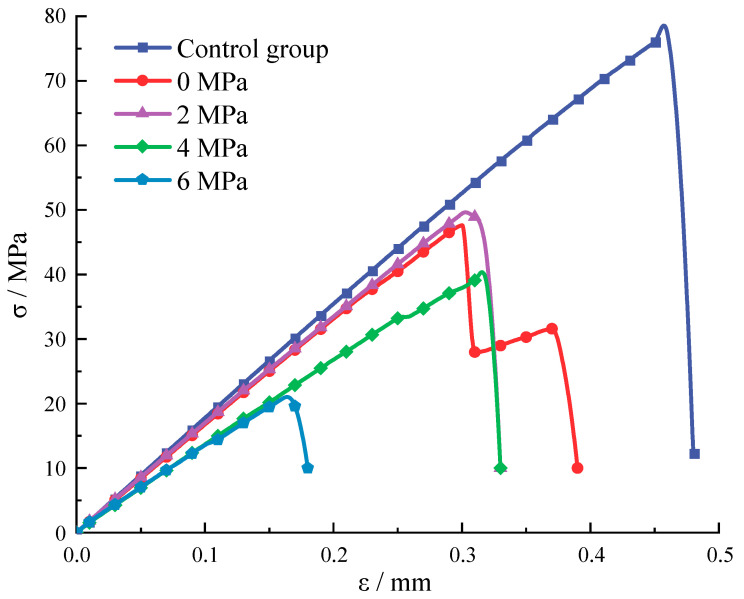
The whole process of stress-strain in rocks containing circular stress inclusions with different stresses.

**Figure 15 materials-16-07519-f015:**
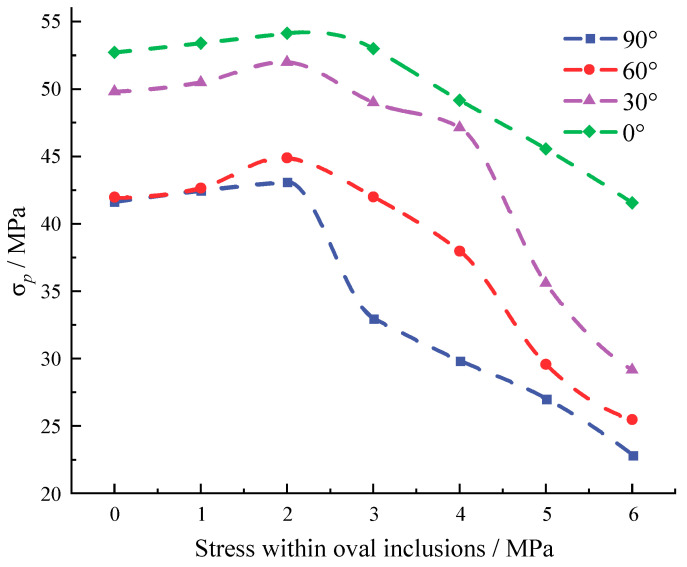
Maximum stress variation of inclusions with different stress ellipsoidal stresses.

**Figure 16 materials-16-07519-f016:**
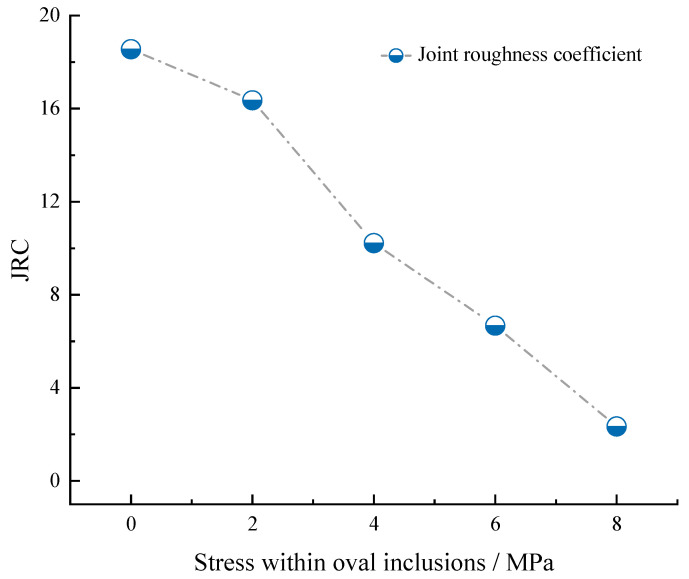
JRC curves for rock failure containing 30° elliptical stress inclusions.

**Figure 17 materials-16-07519-f017:**
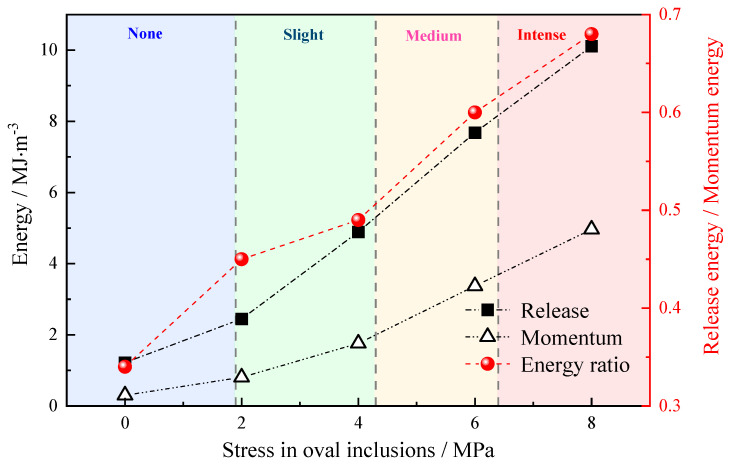
The ratio of failure releases energy to kinetic energy for rocks containing 30° elliptical stress inclusions.

**Figure 18 materials-16-07519-f018:**
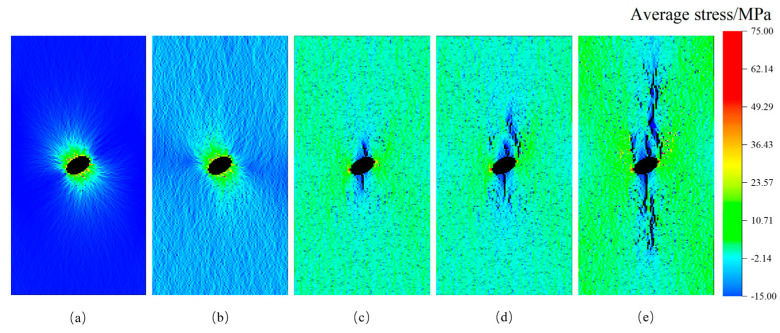
Failure processes in rocks containing 30° stress inclusions under 4 MPa confined stress conditions. (**a**) Initial stage; (**b**) Elastic stage; (**c**) Initial crack stage; (**d**) Microcrack growth stage; (**e**) Microcrack linkage stage.

## Data Availability

Data are contained within the article.

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
