# Peer review of "Numerical Study on the Impact of Locked-In Stress on Rock Failure Processes and Energy Evolutions"

_materials, 2023, doi:10.3390/ma16247519_

Round 1

Reviewer 1 Report

Comments and Suggestions for Authors

General comments:

Although the models presented show some interesting results, the paper in its present form does not clearly explain what the authors have done, describe the results, or present a coherent analysis and discussion of the implications of the work. I would therefore suggest that a major rewrite is required before the paper is ready for publication.

The introduction presents a good review of previous work but does not explain what is the purpose of the current study, present a hypothesis to be tested, or describe how the authors propose to do this. It is not even clear at this stage whether this is an experimental or modelling study (it is in fact a modelling study with no experimental results). Section 2 presents some theoretical background but is not at all clear or easy to follow: diagrams do not relate clearly to text, and terminology is misused or incorrect. It is not clear how the analytical formulae presented here are to be used. Section 3 presents the numerical model results, but the authors do not give any details about the code they have used e.g. what numerical methods does it use?, what is the mesh size?, does it use elastic or inelastic constitutive laws?, etc. The discussion in Section 4 is also difficult to follow, and is often not supported by the model results presented here (although it may be supported by model results that are not presented, this is impossible for the reader to know).

Specific comments:

Line 81: “a circular inclusion of radius a,” Is this a spherical inclusion or a flat disk? It is not clear from the text or the diagrams.

Line 83: “A uniform pressure q1/2 and tension q1/2”: Fig 1 suggests the boundary stress is compressive in all directions.

Line 85: “while the inclusion is not subjected to any confined stress”: Is the inclusion acting as a void (i.e. zero stress on the inclusion surface) or as a rigid inclusion (i.e. 0 strain on the inclusion surface)?

Fig 1b: Is this an axisymmetric 2D model?

Line 133, eq 3: “we can get the analytical solution of the stress value at any point of the demonstration rock with-out closed stress.” But equation 3 gives expressions for the strain (epsilon and gamma) not the stress.

Line 157: “Assume the symmetry axis is the z-axis, perpendicular to the oxy plane”, but in Figs 1-4 the axis of rotational symmetry is the z axis.

Line 165: “the external force is obtained after considering the displacement boundary conditions”, but equation 4 describes stress boundary conditions not displacement.

Line 183: “numerical simulation experiments were conducted in RFPA”: What is RFPA? Is it a numerical code? In any case you need more details about the code and the model set up – numerical methods, mesh sizes, constitutive laws, etc.

Lines 189-193: What is the outer boundary stress (i.e. q) in these models?

Line 210: “rocks without internal stresses but containing circular inclusions”: does this mean rocks containing voids?

Line 222: “Once the locked-in stress within the stress inclusions reached the tensile strength”: What is the tensile strength? In any case there is no obvious difference between the patterns in any of the models presented in Fig 7. In Fig 8, the main change caused by increasing internal stress is the position of fracture nucleation, which moves from the ends of the ellipse (left and right) towards the centre (top and bottom).

Line 245: “and the capacity to heal fractures following fracture initiation”: You do not seem to include fracture healing processes in any of these models.

Line 247: “with varying lamination angles”: The models do not seem to include laminations either.

Line 249: “with initiation cracks typically generated in the middle of the specimen”: What do you mean by initiation cracks?

Line 253: “above the tensile strength threshold”: What is the tensile strength threshold.

Line 322: “pole diameter”: What do you mean by pole diameter? It is not labelled on any of the diagrams.

Line 327: “the study of cloud diagram evolution”: What do you mean by cloud evolution?

Lines 327-339, Figure 15: Description and calculation of the JRC should be in the Results section, not the Discussion.

Lines 340-356: You need a more detailed description of how the dissipated energy and kinetic energy is calculated. Do you assume a value for the crack surface energy? If so what value, and why?

Line 366: “rocks exhibit uniform deformation in the elastic stage, attributed to the closure of initial cracks and deformation between mineral particles (Figure. 7(a) and 8(a))”: Figs 7a and 8a do not show deformation in the elastic stage – they have large cracks along the length of the model, i.e. the post-failure stage.

Lines 366-382: The process of crack growth and progressive rock failure is not shown in any of Figs 7-11, as these only show the final, post-failure stage at different internal pressures. It would be useful to select one or two models and show a sequence of intermediate stages, e.g. the elastic stage, the stage with initial isolated microcracks, the microcrack growth stage, the microcrack linkage stage, etc.

Reviewer 2 Report

Comments and Suggestions for Authors

It's quite good paper. Minor editorial errors are highlighted in gray in the attached file. Figures 2, 4 and 6 are too small and their quality is insufficient.

I am missing a table with the mechanical properties of rocks and how they were determined (laboratory tests?).

Comments on the Quality of English Language

Reviewer 3 Report

Comments and Suggestions for Authors

The central theme of the manuscript involves numerical study on the impact of locked-in stress on rock failure processes and energy evolutions.

Literature discussions and conceptual aspects in terms of the stress distribution of rock fractures and the impact of stress inclusion on the rock deformation and failure are presented in a very in-depth and consistent way.

However, the approach to the experimental numerical simulation protocol is not very clear. It is suggested to the authors that the methodology be more explicit, as well as the results obtained and related discussions.

The authors report, in the summary, that the results obtained in the research provide a solid theoretical basis for ensuring the safety of excavations in various deep engineering projects. 

Such considerations should be better contextualized in terms of the geomechanical aspects studied and perhaps aspects of different geotechnical conditions of geological masses present in deep engineering works.

Round 2

Reviewer 3 Report

Comments and Suggestions for Authors

I understand that the modifications made by the authors took into account the recommendations and suggestions made previously.

Author Response

Thank you for your suggestions and modifications!